# Soil-Related Sustainable Development Goals: Four Concepts to Make Land Degradation Neutrality and Restoration Work

**Saskia Keesstra [1,2,\*], Gerben Mol [3], Jan de Leeuw [4], Joop Okx [1], Co Molenaar [5], Margot de Cleen [5] and Saskia Visser [6]**

1   Soil, Water and Land Use Team, Wageningen Environmental Research, Droevendaalsesteeg 3, 6708RC Wageningen, The Netherlands; joop.okx@wur.nl
2   Civil, Surveying and Environmental Engineering, The University of Newcastle, Callaghan 2308, Australia
3   Climate Change, Water & Food Team, Wageningen Environmental Research, Droevendaalsesteeg 3, 6708RC Wageningen, The Netherlands; gerben.mol@wur.nl
4   International Soil Reference and Information Centre (ISRIC), Droevendaalsesteeg 3, 6708RC Wageningen, The Netherlands; jan.deleeuw@wur.nl
5   The Ministry of Infrastructure and Watermanagement, Rijkswaterstaat, Griffioenlaan 2, 3526LA Utrecht, The Netherlands; co.molenaar@rws.nl (C.M.); margot.de.cleen@rws.nl (M.d.C.)
6   Environmental Sciences; Wageningen University and Research, Droevendaalsesteeg 3, 6708RC Wageningen, The Netherlands; saskia.visser@wur.nl
\*   Correspondence: saskia.keesstra@wur.nl; Tel.: +31-624531520

**Abstract:** In the effort to achieve the Sustainable Development Goals (SDGs) related to food, health, water, and climate, an increase in pressure on land is highly likely. To avoid further land degradation and promote land restoration, multifunctional use of land is needed within the boundaries of the soil-water system. In addition, awareness-raising, a change in stakeholders' attitudes, and a change in economics are essential. The attainment of a balance between the economy, society, and the biosphere calls for a holistic approach. In this paper, we introduce four concepts that we consider to be conducive to realizing LDN in a more integrated way: systems thinking, connectivity, nature-based solutions, and regenerative economics. We illustrate the application of these concepts through three examples in agricultural settings. Systems thinking lies at the base of the three others, stressing feedback loops but also delayed responses. Their simultaneous use will result in more robust solutions, which are sustainable from an environmental, societal, and economic point of view. Solutions also need to take into account the level of scale (global, national, regional, local), stakeholders' interests and culture, and the availability and boundaries of financial and natural capital. Furthermore, sustainable solutions need to embed short-term management in long-term landscape planning. In conclusion, paradigm shifts are needed. First, it is necessary to move from excessive exploitation in combination with environmental protection, to sustainable use and management of the soil-water system. To accomplish this, new business models in robust economic systems are needed based on environmental systems thinking; an approach that integrates environmental, social, and economic interests. Second, it is necessary to shift from a "system follows function" approach towards a "function follows system" one. Only by making the transition towards integrated solutions based on a socio-economical-ecological systems analysis, using concepts such as nature-based solutions, do we stand a chance to achieve Land Degradation Neutrality by 2030. To make these paradigm shifts, awareness-raising in relation to a different type of governance, economy and landscape and land-use planning and management is needed.

**Keywords:** land degradation neutrality; soil-water system; regenerative economy; connectivity; nature-based solutions; land use planning

## 1. Introduction

In 2015, the United Nations (UN) adopted the Sustainable Development Goals (SDGs): a set of 17 goals intended to be a roadmap for society to move from exploitation to the sustainable use of our planet's resources, and from inequality, poverty, and hunger to a proper education and good life for all [1,2]. A robust soil-water system is essential for achieving most of these SDGs. In this paper, we focus on SDG 15.3, Land Degradation Neutrality (LDN) and land restoration, which is part of SDG 15, Life on Land. Ever since the SDGs were adopted by all UN members, how to implement SDG 15.3 has been under discussion. Although there is still discussion among scientists and policymakers about the definition of Land Degradation [3], estimations show that worldwide, 75% of the land is degraded [4]. Figure 1 shows the interdependency between the SDGs. To achieve the SDGs related to food, health, water, and climate, an increase in pressure on land is to be expected. Avoiding further land degradation is therefore a challenge. Not only is the multifunctional use of land and the soil-water system within the system boundaries needed, a change in both stakeholders' attitudes and in economics is essential. Therefore, a change in mind-set or a paradigm shift is inevitable [5].

Achieving a balance between the domains of the economy, society, and the biosphere, as shown in Figure 1, calls for a holistic approach. Solutions depend on the level of scale (global, national, regional, local), stakeholders' interests and culture, and the availability of financial and natural capital. In this paper, we introduce four concepts that we consider to be conducive in realizing LDN. These concepts are partly complementary and reinforce one another: systems thinking, connectivity, nature-based solutions, and regenerative economics. We focus on the contribution and boundaries of the soil-water system.

## 2. Land Degradation Processes

A recent paper by Keesstra et al. [6] highlighted the role of properly functioning soils that can generously provide their ecosystem services for the achievement of the SDGs (Table 1 lists the services and SDGs). In many governance and regulatory documents, the need to conserve this integrated picture of a healthy, properly functioning soil is not reflected. In most (northern) European countries, land degradation is characterized in policy documents with only two aspects: physical degradation focusing on water erosion and landslides, and chemical degradation, mainly focusing on point pollution (e.g., heavy metals, toxic organic compounds) and nitrate and phosphate concentrations. Other soil threats like compaction, soil subsidence, organic matter decline, diffuse pollution, and biodiversity decline, are often not taken into account even though they are abundantly present and seriously affect proper soil functioning, which in turn affects the natural, social, and economic capital the land provides. The land degradation processes in this paper are grouped into three commonly used categories—physical, chemical, and biological degradation—and show whether they are connected to the ecosystem services and the SDGs (Table 2). Most forms of land degradation are touched upon in these three groups, but landslides are left out because they involve large-scale landscape changes and are generally not the result of poor land management. Soil sealing has also been left out because it is not a natural land degradation process, but a form of land use. For each of the categories, we give a brief overview of the processes and the state of degradation and suggest measures to mitigate degradation.

**Table 1.** Links between degradation processes, ecosystem services, and sustainable development goals.

| Land Degradation Processes | | Ecosystem Services (Table 2) | Sustainable Development Goals (SDGs) |
|---|---|---|---|
| Physical processes | Compaction | 1,3,4,5,7,8 | 2,6,12,15 |
| | Erosion | 1,2,3,4,5,6,7,9,10,11, | 1,2,6,12,13,15 |
| Chemical processes | Salinization/acidification | 1,2,4,5,6,7,8,10,11 | 1,2,6,12,15 |
| | Contamination | 1,2,5,7,8,9,10,11 | 1,2,3,12,15 |
| Biological processes | OM decline | 1,4,5,6,7,8 | 1,2,3,12,13,15 |
| | Biodiversity loss | 1,2,4,5,6,7, 8,9,10,11 | 1,2,12,15 |

**Table 2.** Overview of ecosystem services and Sustainable Development Goals (SDGs) relevant for Land Degradation Neutrality.

| Ecosystem Services | | Sustainable Development Goals Relevant for Land Degradation Neutrality | |
|---|---|---|---|
| 1 | Provision of food, wood and fiber | 1 | No poverty |
| 2 | Provision of raw materials | 2 | Zero hunger |
| 3 | Provision of support for infrastructure for humans and animals | 6 | Clean water and sanitation |
| 4 | Flood mitigation | 7 | Affordable and clean energy |
| 5 | Filtering of nutrients and contaminants | 8 | Sustainable economic growth |
| 6 | Carbon storage and greenhouse gases regulation | 9 | Resilient Infrastructure |
| 7 | Detoxification and the recycling of wastes | 11 | Sustainable cities |
| 8 | Regulation of pests and disease populations | 12 | Responsible consumption and production |
| 9 | Recreation | 13 | Climate Action |
| 10 | Aesthetics | 15 | Life on land |
| 11 | Heritage values | | |
| 12 | Cultural identity | | |

## 2.1. Physical Degradation

Physical land degradation involves the displacement and/or repositioning of soil particles without altering their chemical composition. The best-known example of displacement is erosion caused by both wind and water, whereas compaction is an example of the onsite repositioning of soil particles caused, for example, by excessive pressure from heavy agricultural machinery and/or drainage/dewatering. Achieving land degradation neutrality means eliminating net erosion or bringing the erosion rate in equilibrium with the soil formation rate [7]. Erosion results from a complex interplay of processes that ultimately lead to loss of soil fertility [8], loss of organic matter [9], and loss of the top soil that provides water and nutrient holding capacity [8]. Degraded soils slack and crust easily [10] and have lower infiltration rates [11], which creates higher runoff and further erosion [12]. These on-site effects are not the only issues that matter, however. Off-site effects of erosion can be equally damaging. Wetlands and marine ecosystems downhill of the erosion sites can be polluted and smothered by sediment and associated substances [13].

Compaction is a form of land degradation that is particularly important in agricultural soils worked by heavy machinery, leading to compaction of shallow and deeper soil layers. In some cases, trampling by people or livestock causes shallow compaction [14,15]. Compaction due to drainage of clay and peat soils in combination with loss of organic matter leads to soil subsidence with very high costs for society [16].

Erosion and compaction are strongly connected to soil structure, infiltration, and water holding capacity. Therefore, these parameters could assess physical degradation of a soil system, However, infiltration varies over time and space due to local phenomena such as macro-pores [17], water repellency [18], and biota [19].

### 2.2. Chemical Degradation

Chemical degradation of soils is the result of multiple processes, including the (over)use of manure and fertilizers (nitrogen and phosphorus) leading to eutrophication of soils and ground- and surface waters through leaching and runoff, pollution by inorganic (e.g., heavy metals, radionuclides) and organic substances (e.g., insecticides, herbicides, PCB, PAH), and salinization in (semi-)arid regions of the world. This paper does not extensively review all these processes, but briefly describes their essential characteristics.

Chemical degradation of soils as a result of heavy metals, radionuclides, and organic substances is a widespread form of diffuse pollution, which affects the biotic and abiotic functioning of the soil, the quality of crops, and the health of animals and humans. The main soil processes determining the availability both for uptake by plants or animals as well as for leaching of heavy metals and radionuclides are sorption, complexation, and precipitation. Together with soil pH and redox state, these processes determine the amount of each metal in the soil solution, which is the (bio)available part. Most metals tend to be more available at low pH (acid soils). Radionuclides mostly show a similar geochemical behavior, only with the added risk of radiation. Organic pollutants show different behavior in many ways; two important reasons being that they can potentially decompose (so-called natural attenuation), and that they have in many instances been designed to be biologically active molecules. The first aspect is often hard, hence the term persistent organic pollutant (POP), but when it happens it sometimes results in metabolites that are more bioavailable and more toxic than their parent molecules [20]. The second aspect is exemplified by biologically active compounds like insecticides and herbicides, but lately also by the introduction into the environment of substances related to veterinary and human medicine and hormone use and dumps of illegal drug laboratories.

The third important chemical soil degradation process is salinization due to inadequate water management and/or climate change in delta areas. This form of soil degradation is widespread in semi-arid parts of the world [21]. The degradation is the result of inappropriate use of irrigation without sufficient drainage over a long period of time (water management) or excessive drainage (polders). Salinization puts the long-term use of irrigated agriculture, which is important for food security (SDG 2) and enough drinking water (SDG 6), at risk in the long term. Due to climate change, salinization might become a bigger problem in some areas and decrease in others.

### 2.3. Biological Degradation of Soil Organic Matter

Soil organic matter (SOM) is reputed to enhance almost all aspects of soil functioning, ranging from soil fertility to soil structure, from water retention and infiltration capacity to the regulation of nutrients, and from prerequisites for a rich soil ecosystem to a carbon pool of global importance [22–28]. The largest anthropogenic influence on SOM is the conversion of native forests and grassland to arable land [28,29]. Processes controlling the SOM balance are the rate of addition of C in the form of plant residue, manure and other organic waste, for example, and the rate of loss due to decomposition. Processes that determine the stabilization or decomposition of SOM are thought to be less determined by the molecular structure of the organic matter but are much more dependent on the environmental and ecological circumstances [25].

From this perspective of SOM dynamics, the diversity of soil organisms is very important, both macro fauna like beetles and earthworms and microbiota like fungi and bacteria, which can increase the stabilization of SOM as well as speed up its decomposition [29]. The general idea, however, is that a rich soil ecosystem enhances the structural connection between the mineral and organic phases in the soil, and that this mineral-stabilized soil carbon is responsible for long SOM residence times and increased SOM levels, especially in the deeper soil layers [30]. The diversity of soil organisms is crucial to many functional aspects of soils such as soil structure and water regulation, and nutrient cycling (there is less need for artificial nitrogen fertilizer), but especially also to the C-dynamics of soils [29,31]. Global patterns of soil biodiversity loss are also reflected in the soil. A study by Tsiafouli et al. [31] showed that all over Europe soil biodiversity is in consistent decline under the pressure of intensive

agriculture. This consistent decline in all soils of Europe, no matter the soil type or agricultural use, results in a loss of ecosystem services that can be obtained from the soils [32].

To ensure a good quality SOM, it is probably easier to ensure proper soil biodiversity; a rich soil ecosystem almost always improves the quality of SOM, its interaction with the mineral parts of the soil, and the ecosystem functioning of soils [33].

## 3. Socio-Economic and Policy Processes

Currently, the cost of land degradation reaches about €420 billion per year, much higher than the cost of action to prevent it [34]. To effectively tackle land degradation, its drivers should be addressed, and instruments designed to incentivize the sustainable management of land. Embedded in the understanding of the 'economics of land degradation' is a set of methodologies for assessing the true societal impacts of land degradation. These form the cornerstone for determining how to best allocate financial, technical, and human resources to tackle land degradation [34].

The drivers with a big impact on land degradation are: increasing world population, human aging, urbanization, climate change, growing welfare and increasing (protein) consumption, growing pressure on natural capital and resources, and growing energy consumption. All these drivers interfere with one another.

The impact of these drivers can be influenced by socio-economic processes and policy. Land degradation neutrality (LDN), as promoted by the United Nations Convention to Combat Desertification (UNCCD) and partners, has found entry into the Sustainable Development Goals (SDGs), and it is hoped that alongside an agreed definition and indicator framework, it will remain part of the framework of development agendas [35]. Caspari et al. [35] also emphasize that the process would be facilitated if the following conditions were met:

- Awareness about soils and land: people living off the land tend to have a strong drive to protect and sustainably manage their land assets; efforts have to be increased to raise awareness of the vital functions of land and soil and the destructive consequences of the "cost of inaction" at the level of politicians and decision-makers.
- Respect for complexity of the subject: the various existing assessments—and especially their incompatibility with each other—have proven that there is no simple solution to the issue.
- Acceptance of the ecosystem approach: all the relevant services that land and soil provide need to be considered.
- Respect for cultural diversity: any global assessment must be open to local interpretations of land and soil quality and be allowed to define and use its own ranking system.
- Science-policy interface and lobby for soils: one of its primary tasks would be to provide a globally-accepted framework for land and soil assessment.

Since the majority of land degradation processes takes place on private property, private efforts next to public efforts are necessary for reaching the sustainable development goals. Thus, initiatives for a circular economy [36] or civil participation can be considered as drivers, as well as private investments and subsidies in sustainable energy and civil desires for healthy food and nature perception and a future for the next generation. The Ecosystems and Landscape Management Cluster of the World Business Council for Sustainable Development (WBCSD) has engaged in a discussion around LDN by establishing the Land Degradation Neutrality for Business initiative. The ultimate purpose of the initiative is to engage business in the process of translating the LDN concept into concrete action and to clarify the business contribution to the LDN target [37]. This publication calls for a number of actions by public and private decision-makers that can help business further engage in scaling up sustainable land management, restoration, and rehabilitation and, subsequently:

- Explore and pilot smart, scalable LDN business models that secure returns on investment, building on existing profitable and sustainable practices.

- Assess the cost of land degradation and the total economic benefits of LDN and factor them into sustainable land management, restoration, and rehabilitation activities to become maintenance practices that are needed to preserve the value and function of land, just as for any other company asset.
- Increase awareness of the cost of inaction and the benefits of action across the value chain to encourage producers and consumers to change their current production and consumption patterns towards more sustainable ones.
- Support business action through long-term, smart, measurable policies, regulations, and incentives that provide a level playing field.
- Put in place enabling conditions to allow small-scale producers to engage.
- Turn governments and companies into brokers, helping facilitate dialogue and partnerships to ensure the fair distribution of costs and benefits arising from sustainable land management and land restoration.

This shows that to combat land degradation and stop land degradation processes, not only is insight into physical, chemical, and biological processes needed (as indicated by the arrow connecting the layers in Figure 1), but knowledge about autonomic processes driven by changes in the economy and society is also required. Policy instruments that influence these processes on different levels of scale are needed, as well as new business models and stakeholder involvement. This complex of processes and involvement calls for a holistic approach. This should lead to a joined vision and strategy.

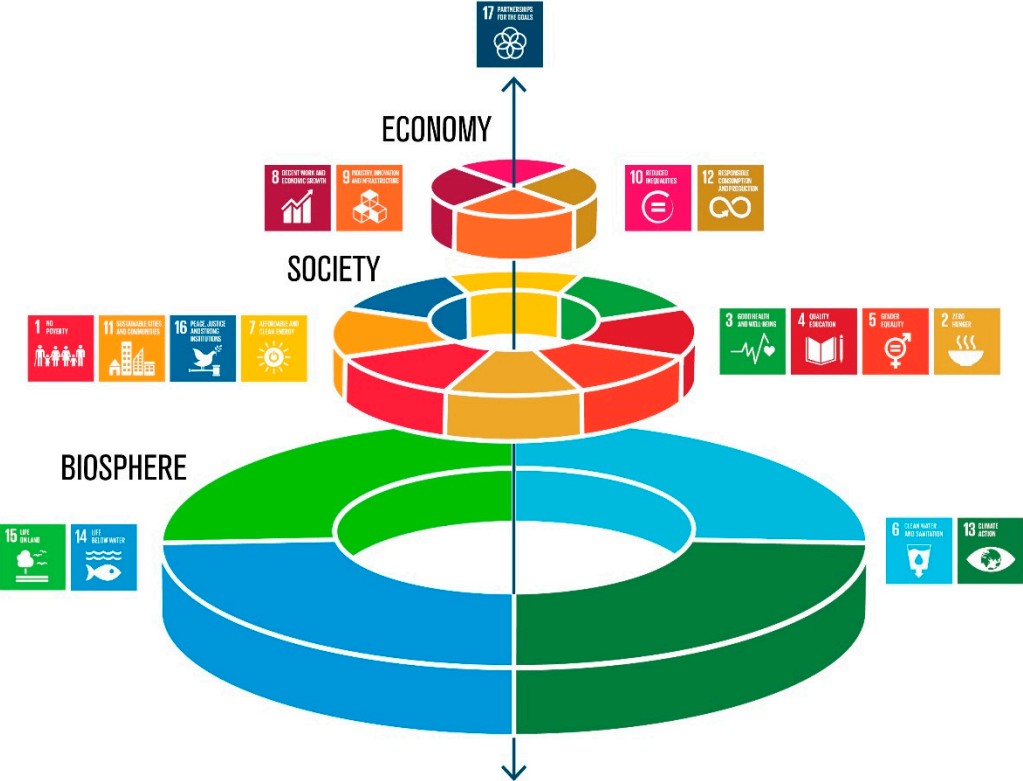

**Figure 1.** Relation of different domains within the Sustainable Development Goals (SDGs), biosphere, society, and economy. SDG15 is one of the four goals making up the biosphere ring (adapted after the original figure of the Azote Images for Stockholm Resilience Centre).

## 4. Four Concepts to Make LDN Work

To achieve LDN, the ideal situation is a holistic approach that incorporates the biosphere, society, and the economy as well as the driving processes. As we do not have all the competences, we have

restricted ourselves in this paper to identifying the essentialities in the field of the biosphere as part of a holistic approach. At present, policy decisions are mainly economy-driven. To enhance the incorporation of the biosphere, awareness-raising, agreement about the value of the soil-water system, and knowledge and data about the status, boundaries and response of the system, are essential. A complicating factor is that degradation takes place on different spatial scales, which often exceed governance boundaries, while the decisions are made on governance boundaries. These boundaries are often misleading. Authorities should be aware that to optimize solutions for SDGs by incorporating the services of the soil-water system, they need to collaborate. We introduce four concepts, each concerning different levels of scale that support holistic decision-making: (i) systems thinking; (ii) connectivity; (iii) nature-based solutions; and (iv) regenerative economics. These concepts are partly complementary. Concept (i), systems thinking, is introduced because it is essential to understand the impact of land management and land use changes to optimize the sustainable use of the system. Concept (ii), connectivity, is chosen because it shows not only the impact of land management and land use changes within one natural system, but the impact of natural systems which are spatially or geographically connected and influence one another, such as river basins and groundwater bodies. Concept (iii), nature-based solutions, uses the characteristics and dynamics of the natural system to enhance resilient solutions. Concept (iv), regenerative economics, is of a slightly different order. At present, societal and environmental costs are insufficiently taken into account in planning and decision-making. A growing awareness exists that to achieve the SDGs, changes in economics are essential. Therefore, the challenge is to value of the natural system correctly. Other concepts with a bigger focus on the social and economic appraisal, like Land Stewardship, life cycle analyses, and the donut theory, are not discussed in this paper.

### 4.1. Systems Thinking and the Heterogeneity and Dynamics of Land and Soil

Landscapes and ecosystems are complex systems consisting of interacting subsystems of a geological, climatological, ecological, and also human nature, among others. Getting a handle on the behavior of complex systems is generally achieved through the use of three concepts: stocks and flows, feedback loops, and delayed response [38]. A stock of something valuable like fertile soil is influenced by its 'outflow', for example, in the form of erosion and its 'inflow', e.g., in the form of soil formation. Feedback loops within and between subsystems provide the functional connections in the system in two fundamental ways: reinforcing feedback loops, and balancing feedback loops. Reinforcing feedback loops amplify the effect; for example, loss of vegetation on a slope leads to increased erosion, which removes fertile soil which in turn leads to further loss of vegetation. Balancing feedback loops make a system resilient because they lead the system back to equilibrium when it is subjected to a disturbance [39–42]. An example is the presence of shrubs on arctic soils which trap snow, leading to a higher soil temperature than in bare soils. This higher soil temperature leads to a higher activity of the microbial soil community, in turn resulting in a higher amount of mineralized N for the shrubs [43].

Sometimes these feedback loops kick in with a delayed response, a phenomenon that occurs particularly in systems with a large buffer capacity like soil. Such buffering subsystems are stable for a long time, taking up the disturbance until a critical threshold is reached and the subsystem changes to a fundamentally different regime associated with entirely different feedback loops. A well-known example is the eutrophication of groundwater due to nutrient leaching after decades of over-manuring of the soils in the Netherlands.

In the words of Raworth (2017, [44]), systems emerging out of these interactions between stocks, flows, feedbacks, and delays are "complex because of their unpredictable emergent behavior, and adaptive because they keep evolving over time". Indeed, natural landscapes and ecosystems are usually in some sort of dynamic equilibrium state, where balancing feedbacks bring them back to equilibrium when the system endures pressure from an external driver. Moreover, they are adapted to the natural setting in which they have developed, such as the climate, geology, hydrology, soil type, and, in most cases nowadays, also the land management implemented in the area. The increasing

pressures on land because of competing claims and unsustainable modes of soil and land management push these dynamic systems out of equilibrium into a state of disequilibrium where they show transient behavior that is often hard to understand.

Finding solutions in such complex systems calls for a sophisticated understanding of the (non-linear) mechanisms that make them work, but also of where to intervene in the system. Meadows (2009, [38]) identifies 12 so-called leverage points that can be used to influence any system, ranging from leverage points that have a low impact like subsidies, taxes, and environmental standards on the one hand via information flows and rules (incentives, punishments, and constraints), to complete paradigm shifts when goals of communities or society change. Subsidies, taxes, and environmental standards have a low impact because they almost never influence how the system works, but only correct an unwanted symptom. In the middle of the leverage points scale is the flow of information; having access to information on what your management choices imply helps with making the most sustainable choices. Meadows gives a simple example from the Netherlands of electric meters being placed in the basement of some houses and in the front hall of other houses. This led to a 30% reduction in electricity consumption in the houses where the information was readily available in the front hall. At the top end of intervening power are aspects like paradigm changes and the resultant goal setting of the system (a company changing its goal from maximizing profit with food production to making SDG 2 Zero Hunger occur at a profit margin that keeps the company healthy).

Land and soils are complex heterogeneous and dynamic systems. They evolved as the result of biophysical processes in the past that frequently still operate today. Our understanding of the heterogeneity and dynamics of land and soils developed through a combination of empirical research and models that allowed us to understand the implications of these past processes over space and time (4D). Computer-based systems analysis has broadened and deepened our understanding of the influence of feedbacks on system behavior and the importance of emergent properties such as resilience and stability and self-organization as a result. Van den Koppel et al. (2005, [45]), for example, showed that the positive feedback between vegetation and sediment deposition induces uplift, self-organization, and resilience of salt marsh systems. This reduces the extremes of the intertidal environment and promotes the growth of plants that trap sediment. At the same time, uplift of the marsh surface through sediment deposition drives salt marshes to a critical state where steepness of the marsh tidal-flat edge increases vulnerability to wave-induced erosion. Such knowledge of the soil-water processes is essential for decision-making. Showing the boundaries of the system helps to avoid land degradation and prevents the transfer of damage to other ecosystems or stakeholders [46].

*4.2. Connectivity*

The connectedness between systems in a spatial or geographical way is, in general terms, referred to as connectivity. The connectivity concept can thus be considered as a special case of systems thinking related to the spatial realm. Because functional connectedness in space is especially important when land use is concerned, we present connectivity as a separate concept. The connectivity concept allows us to consider the off-site effects of systems, e.g., plastic pollution in oceans, downstream effects of upstream changes in water, and sediment balance. Thus, this concept is especially useful for (broader) area management. Figure 2 gives an overview of the processes that impact water and sediment connectivity on the catchment scale.

The connectivity concept entered the physical geography research realm relatively recently. In a COST Action on this topic (COST Action ES1306) progress has been made towards a general conceptualization and implementation of this concept in land management. The concept has its base in a holistic approach to catchment systems like the Delta's, in which multiple drivers (e.g., increased rainfall due to climate change, or changing land management practices due to diminishing economic returns) impact the processes and feedbacks [47–49]. An understanding of the impact of these drivers and their geographical interaction enables land managers to adjust the management of the system in the direction of long-term sustainability.

The concept of connectivity demonstrates the need to seriously consider where to place the boundaries of the area, which subsystems in the catchment or other geographical unit under consideration comprise the whole system, or how we want to consider the effects of on-site management interventions (within the system boundaries) on off-site locations (outside the system boundaries).

A classic study by Trimble (1999, [50]) in a reforesting catchment in the USA showed that the sediment load delivered at the outlet of the catchment remained the same while the sediment production in the uplands was reduced by 90%. However, through the reworking of the sediment stored in the valley bottoms, the sediment yield at the outlet remained the same. This illustrates the complexities involved in systems and their connectivity, and that intricate knowledge of the water and sediment dynamics is essential to realize land degradation neutrality. In land management, it is therefore important to take off-site effects of on-site processes into account when deciding on a specific intervention. Good knowledge of the connectivity of the soil-sediment-water system is essential to make this assessment.

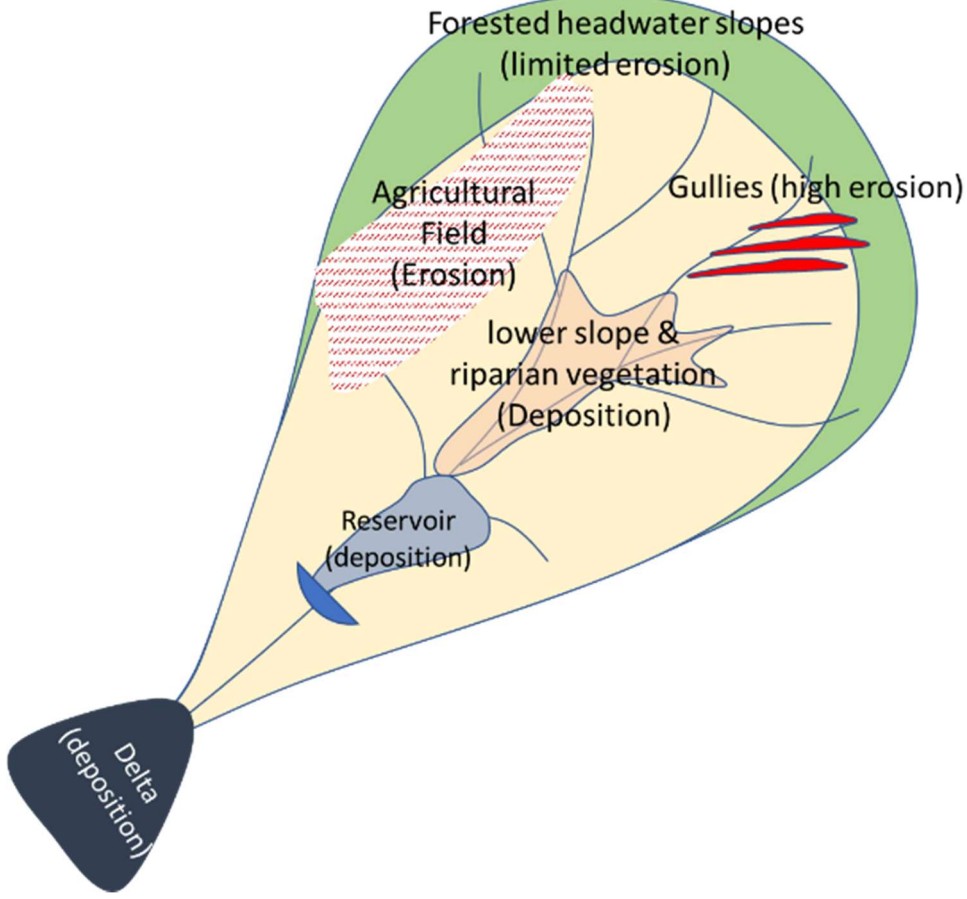

**Figure 2.** Illustration of sediment connectivity within a catchment.

### 4.3. Nature-Based Solutions

The concept of Nature-Based Solutions (NBS) is everywhere nowadays. It can therefore be considered as an umbrella concept comprising a large variety of approaches to ecosystem restoration, climate adaptive infrastructure (building with nature), and disaster risk reduction [51]. The EU Research and Innovation policy agenda on NBS defines them as solutions that are "inspired and supported by nature, which are cost-effective, simultaneously provide environmental, social and economic benefits and help build resilience. Such solutions bring more, and more diverse, nature and natural features and processes into cities, landscapes and seascapes, through locally adapted,

resource-efficient and systemic interventions" (https://ec.europa.eu/research/environment/index.cfm?pg=nbs).

The term Nature-Based Solution emerged about ten years ago with publications by MacKinnon and colleagues [52,53] and is strongly related to terms like Ecosystem Services, natural capital, ecological engineering, and blue-green infrastructure, etc. [54]. Central to most of these concepts is the fact that they seek to make use of natural alternatives to purely technological solutions. They do so to solve a whole range of problems, from the maintenance of biodiversity and the restoration of ecosystems, to the design of solutions to cope with climate change and to produce food sustainably [54]. The Sand Engine [55] and the 'Room for the River' [56] projects are good examples of the success of the implementation of this concept as well as natural attenuation as part of risk reduction measures in the management of groundwater pollution [57]. Some essential aspects to consider when designing NBS are: the level of intervention and its 'naturalness' (from minimal interventions to designing artificial ecosystems), the scale of the intervention (from plot scale to whole landscapes), and the complexities this brings in both the natural and socio-economic realms, proper stakeholder engagement, trade-offs, multi- and transdisciplinary knowledge, and mutual learning [54,58,59].

Nature-based solutions in land and water management can be divided into two categories [60]; soil-water system solutions, and landscape solutions. Soil-water system solutions relate to water management and management actions related to a healthier soil system. The soil is promoted to infiltrate better and produce less runoff, to have more organic matter that ensures better hydrological soil functioning and water availability [61,62]. Better soil functioning helps to mitigate floods and droughts, provide better ecosystem services such as higher biodiversity [63], carbon sequestration [28] and nutrient availability [64], and helps to protect groundwater from contamination [64–66]

The second group of nature-based solutions, landscape solutions, have their base in the connectivity concept. These solutions generally provoke dis-connectivity, which helps to reduce the transport capacity of water. Examples can be found in the use of buffer strips on different scales [67], riparian vegetation [68], or wetlands [69]. In landscape planning, both types of solutions should be implemented and combined in such a way that they will enhance each other. As an example, a catchment subject to severe erosion could be treated on-site with soil and water conservation practices, which would promote infiltration and reduce erosion. The runoff should not be routed over roads, but as diffusely as possible, to reduce the transport capacity of the water. The key here lies in keeping the transport capacity low in the transfer paths of the water and sediment, which can be done using grass strips, wetlands, and riparian vegetation on key positions in the landscape.

## 4.4. Regenerative Economics

Two important shortcomings of the current dominant economy are the ignorance of external costs (e.g., to the environment or to the poorest in society) and the linear approach to the use of resources. This leads to an economy that is degenerative and depletes the world's resources. Despite the idea of 'green growth'—decoupling resource use from economic growth—in 2016 the UN reported that the material footprints of almost all countries are still growing fast [70]. Economists and politicians realize this, but so far, they have only tried to correct these market shortcomings with instruments like taxes, quotas, and tiered pricing. Furthermore, although they can be successful to some extent, Raworth (2017, [44]) argues they fall short both in practice and in theory. In practice this is because these instruments are never set at the level that is required due to heavy corporate lobbying and governments being afraid of losing competitiveness. In theory, it is because in systems theory stocks and flows are low-level leverage points to provoke system change [38].

Therefore, alternative economic theories and models are needed. We adopt Kate Raworth's term Regenerative Economics because it encompasses the focus on materials, energy, social capital, and knowledge and ideas, and values them in a more complete way than just from a monetary perspective. Raworth [71] vividly describes the difference in mind-set between degenerative and regenerative economics: "While regenerative designers now ask themselves, 'how many diverse benefits can we

layer into this [activity/design/product/service]?', main stream business still asks itself, 'how much financial value can we extract from this [activity/design/product/service]?'". She calls on economists to come up with metrics that value the regenerative success and not merely financial success.

Other people are voicing similar solutions from different perspectives. For instance, the Capital Institute promotes the concept of Regenerative Capital (see http://capitalinstitute.org/regenerative-capitalism/), aiming to develop financial institutions like banks and investment companies in the direction of less risk and more social involvement in a still profitable way. Mazzucato [72] always stresses the fact that governments have a crucial role to play because only they could have the patience to invest in very long-term innovative directions and, as a result, force breakthroughs. Thus, awareness about the need for a regenerative economy exists, but it is still far from the mainstream economy. In this paper, we stress the need to incorporate the value of the soil-water system and its ecosystem services into policy. These services are often not used as much as they could be, and therefore the potential public benefits are not claimed [73]. The Intergovernmental Science-Policy Platform on Biodiversity and Ecosystem Services (IPBES) on Food [74] also emphasizes the need for a change in mind-set in the food system and to incorporate external costs such as groundwater pollution, land and water use, and landscape degradation.

To value the ecosystem services depends on supply and demand in a certain area. Robinson et al. [75] describe a framework for the state and change of social and natural capital in Europe and are thus able to show degradation in time on a large scale. For decision-making on a regional or local scale, more detailed information is essential as well as a long-term vision, including public and private goods, so that an integrated assessment can be made. To value the services, it should be clear what potential services are available in the area (resource), what interests are at stake (scarcity), and whether use is short- or long-term (time). Therefore, not only are system data and knowledge about different spatial and time scales needed, but also the stakeholders and societal goals (SDGs) in the related area. Instruments like Natural Capital Accounting [76] and the Economics of Ecosystem and Biodiversity [77] have proven to be valuable for the industry and policymakers. The lack of good data, however, can compromise the usability of these instruments.

## 5. Using the Four Concepts to Design Land Degradation Neutral Solutions

SDG 15.3 is formulated as follows: 'By 2030, combat desertification, restore degraded land and soil, including land affected by desertification, drought and floods, and strive to achieve a land degradation-neutral world'. To achieve this target of Land Degradation Neutrality, urgent and integrated action is needed to transition land management in a direction that is environmentally, economically, and socially viable. In this section, we will illustrate how the four concepts described in the previous section can be used to come up with solutions that lead to LDN alternatives for the current situations, and at the same time contribute to other SDGs. We will do so by discussing a few examples and analyzing which concepts they use to be successful; we will then extract general conclusions and recommendations from these cases.

**Example 1.** *Orchards in Eastern Spain, how the system becomes dysfunctional.*

This example shows the opposite trend LDN seeks to achieve. In Eastern Spain and in many other Western Mediterranean countries, the millennia-old tillage practices and irrigation have changed based on new available technology and subsidies. In many Mediterranean countries, the landscape is vulnerable to land degradation due to sparse semi-arid vegetation and a combination of steep relief and highly erosive rainfall intensities. The traditional tillage system depleted the organic matter content but was sustainable from an erosion and water resources point of view. On the hillslopes, there was rainfed agriculture on terraces; in the valleys, there was flood irrigation coming from natural springs. The introduction of drip irrigation using pumped groundwater, also in formally rainfed areas, caused a severe depletion of the groundwater and destruction of hillslope terraces. This is causing large-scale

soil erosion on these transformed hillslopes [78]. In addition, the cultural heritage and social bonding that was based on water management is being lost. This transition is caused by the new drip irrigation technology in combination with European and national subsidies that promote the conversion to drip irrigation and promote no-tillage that persuaded many farmers to use herbicides instead of ploughing their fields, which not only increases the sediment yield [79], but also pollutes the soil and ground and surface water [80]. In a study in Eastern Spain, a herbicide treated no-tillage apricot orchard was found to generate more sediment than ploughed fields. The sediment concentration in the runoff was higher in the ploughed fields, but the runoff was so much higher that the total soil loss in the herbicide treated fields was higher [79].

Placing this example in the framework presented in Figure 3, it becomes obvious what went wrong. The chosen land management only focusses on one thing: making more profit in the short term. The land management strategy does not make use of the soil functions and damages the physical, chemical, or biological processes in the soil. The industrial farming on large plots does not use the concepts we introduced in Figure 3. The large plots enhance the connectivity of water and sediment, facilitating large-scale erosion. Furthermore, the engineered farming system based on fertilizer and herbicide use does not consider the forces of nature that can be used to sustainably farm. When we want to turn around this type of farming, options for sustainable nature-based land management need to use process knowledge embedded in the concept of connectivity. At the same time, however, the economic concepts we introduced must provide the incentive to make more sustainable land management an economically and socially viable option (farmers motivations [81]).

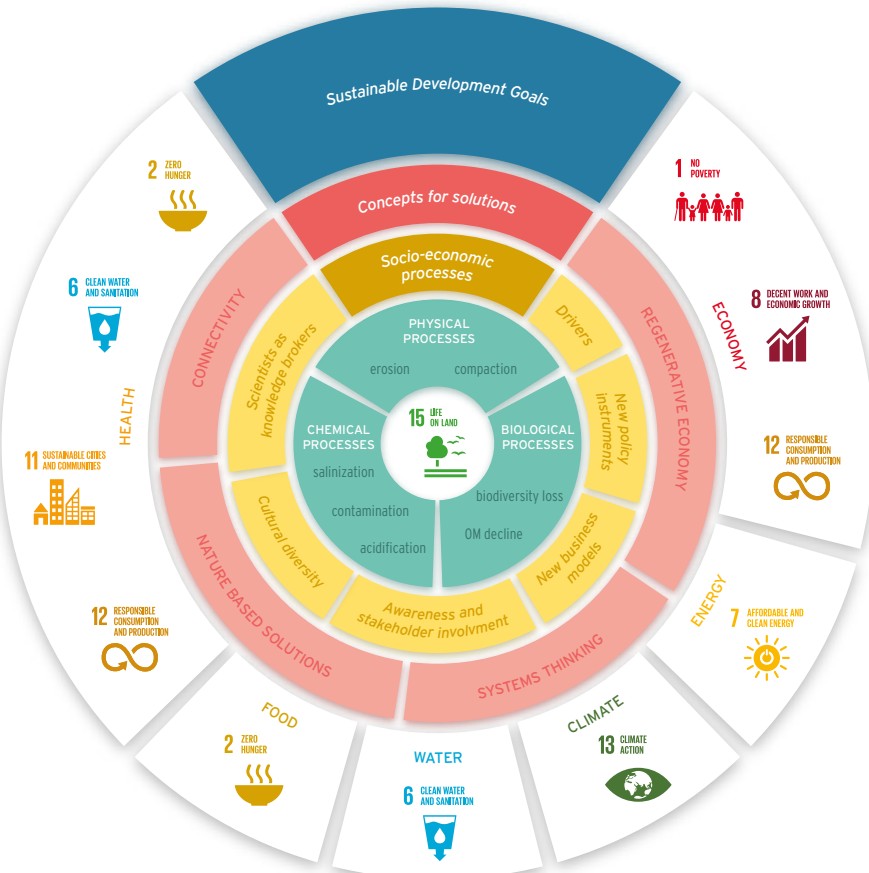

**Figure 3.** Schematic view of the linkages between the Sustainable Development Goals, societal challenges, concepts in soil, land and water management, and indicators and processes in the biosphere (biological, chemical, and physical processes).

This example shows that knowledge about degradation processes and social and economic structures in a specific area is essential to determine the right management, which can not only address land degradation issue (SDG15), but also water quality (SDG 6), responsible production (SDG 11), and sustainable economic growth (SDG 8).

**Example 2.** *The Four returns approach of Commonland.*

Commonland is a non-governmental organization (NGO) working on Land Restoration. Their philosophy is based on a special approach: the four returns approach. These four returns are: return of inspiration; return of social capital; return of natural capital; and return on investment (www.commonland.com). Four returns landscape development projects are long-term, typically lasting for 20 years. They involve local companies' and farmers' initiatives working together on sustainable business. In their project in Altiplano Estepario (a region of 630,000 ha. in Southern Spain), Commonland works on dryland landscape restoration with a community of 250 local stakeholders (https://www.commonland.com/en/projects/187/altiplano-estepario). The most prominent degradation process in this landscape is soil erosion, caused by degrading farming systems that keep the soil bare, except for the crops. The fertile top soil had been lost due to erosion and the introduction of herbicides had caused the fields to become deserts from a biodiversity point of view. The eroding land was causing a range of problems such as loss of soil fertility and agricultural droughts, which led to decreasing agricultural yields.

With the four returns approach, they found ways to tackle the process of soil erosion at the same time as other problems in the area, loss of agricultural production, loss of cultural heritage, loss of water resources, and loss of livelihood. In this case, they found an integrated productive system combining almond orchards with the production of sheep. The soil underneath the almonds is covered with herbs and grasses that the sheep can consume. This fertilizes the soil, sequesters carbon, protects the soil from erosion, enhances the water quality, and is good for biodiversity. In addition, as a second sustainable business case, aromatic plants are introduced. These new sustainable ways of agriculture inspire younger farmers to start a business and form a new community of farmers who combine old and new technologies. This case shows how a solution for land degradation (SDG15) can also address other SDGs (Figure 3), namely SDG 6 (Clean Water and sanitation), SDG 8 (Decent Work and Economic Growth), SDG 12 (Responsible Consumption and Production), and SDG 13 (Climate Action).

Why is this approach a success? Because it is embedded in the people that live on the land, in an approach that can be called the socio-ecological approach. The return of inspiration and the return on investment are based on all four approaches, as described in Figure 3; and because it is an approach that regards the landscape as a multi-faceted system (system thinking). It integrates knowledge about the physical environment (connectivity) and the threats to it with the economic opportunities and the enthusiasm and willingness of the people in the area. It looks for sustainable solutions that work for both the environment (nature-based solutions) as well as for the socio-economic situation (regenerative economy). Without the return of all four capitals, the Commonland approach would not work. However, the implementation of their projects could benefit from better knowledge about the physical environment, about how land degradation processes work, such as the connectivity of the system in terms of water and sediment fluxes, in terms of recovery potential, and the tipping points in a system [82]. In addition, the assessment of the state and the change of the state of the landscape and ecosystem is not well-embedded in the Commonland program, which sometimes causes the unexpected failure of the projects implemented. A better knowledge base for the steps that need to be taken, as introduced in Figure 3, may help ensure the success of these projects. Specific indicators need to be identified and monitored to get a grip on the progress.

**Example 3.** *Circular Economy, Stakeholder Connections and Soil Quality Improvement.*

In a case study in Flevoland, the Netherlands, there is a concern about declining soil quality and fertility partly caused by loss of SOM combined with spatially variable soil setting of subsidence causing fields to have uneven yields. This decline can be partly explained by the use of heavy machinery and deep ploughing, but another important factor is the implementation of short land tenure implemented since 2007. This governance change led to high prices of soil and uncertainty, which causes farmers to opt for crop rotation plans that give short-term high economic returns, but deplete the soil quickly, and damage the soil quality in the long run. This example shows how land tenure decisions made at the governmental level cause land degradation because they add economic stress to the system and thus lead farmers to choose unsustainable options.

Thinking in terms of the whole system concerned and with a focus on regenerative economics would help find proper solutions for this kind of negative spiral. Stakeholders in this region, such as the Veldleeuwerik Foundation and the waste treatment company Orgaworld, are working on these kinds of solutions. The Veldleeuwerik Foundation has several hundreds of associated farmers and works with them on making their agricultural production environmentally-friendly and more profitable from an economic point of view. They do so by helping farmers to devise sustainability plans addressing ten indicators (ranging from soil health to pesticide and energy use and servicing local economies), by organizing companies and other partners in a trustworthy food production chain, and by organizing regional stakeholder meetings. By entering a continuous process of interaction between all stakeholders on the sustainability of the entire food production chain, Veldleeuwerik functions as an agent that brings about change in the (agroproduction) system.

Another stakeholder in the region trying to contribute to this system change is the company Orgaworld. One of their primary activities is closing the regional loop of organic materials. They collect food and garden waste and process it into a range of products that are largely aimed at improving soil quality. The remainder of the organic waste is used for generating green electricity. In this way, these actors in Flevoland work to change the system in a sustainable direction.

The primary goal of the Veldleeuwerik Foundation and Orgaworld is sustainable production (SDG 12). However, their approach also works on SDG 2 (food production), clean water (SDG 6) by using as little fertilizer and pesticides as possible, and SDG 8 (economic growth). These companies have a higher goal with their enterprise than just making money, and they are not waiting for the government to act. In the way they work, the approaches as defined in Figure 3 are largely embedded. Their perceptions of the system (system thinking), working with nature (building with nature), and of creating a regenerative, local economy (regenerative economy) inspire them to make this effort. To truly report on the success of this approach, however, better knowledge, information, and data should be collected on the effects it has on soil quality, food production, farmer income, company profits, etc.

## 6. Paradigm Shifts: Transitions to Economically Viable Sustainability

The case studies show that by integrating the challenges of the SDGs, we stand a chance to achieve them by 2030. The functions of the soil-water system can be used to give guidance to the transitions needed (Table 3). We need to move from 'excessive exploitation' by private parties as well as public organizations or 'full protection' by the government to the 'sustainable use and management' of natural systems and their services through stakeholder cooperation and participation providing public values. This gives a higher regenerative value to the natural system, which creates chances to finance sustainable management of the natural system. Not all land functions have to be realized on all locations within a landscape, but the suitable functions should be implemented tailored to the social, ecological, and economic needs of the communities. This paradigm shift is based on a change in the way in which we approach the natural system. In the past, the system was forced to follow the desired function. This management has driven our soils into a poor state: polluted, compacted, and unsuitable for cropping. According to the alternative paradigm, the function should follow the system's possibilities, in which the boundary conditions for sustainable use of the system lead the permissible land use and management options.

| Concepts | Case 1 | Case 2 | Case 3 |
|---|---|---|---|
| **Systems Thinking** | - | + | + |
| **Connectivity** | - | + | + |
| **Nature-Based Solutions** | - | + | + |
| **Regenerative Economics** | - | + | + |

In each of the case studies, we have illustrated the approach we are trying to advocate. All the examples, albeit with different levels of scale, have a clear land degradation problem linked to the complexities of the economic, social, ecological, and cultural setting. Examples 1 and 2, in particular, both situated in Spain, illustrate that the application of the concepts we introduce in this paper can lead to solutions on both the local and the regional scale. Within this setting, a new system dynamic is being developed together with stakeholders to come to a situation of land degradation neutrality. A long-term vision with goals and long-term stakeholder commitment is a requirement. This approach takes a helicopter view and zooms out from the land degradation issue and tries to identify new business models that will create opportunities to simultaneously work towards other SDGs, and thus improves both the environmental issues as well as the social and economic issues in the area.

To implement the long-term vision, we suggest seeking solutions that take the concepts we have introduced in this paper into account: systems thinking, connectivity, nature-based solutions, and the regenerative economy. These four concepts are strongly interrelated and partly complementary. Their simultaneous use will result in more robust solutions. Systems thinking lies at the base of the three others, stressing not only feedback loops but also delayed responses. An understanding of connectivity helps to arrive at sustainable nature-based solutions. Regenerative economics are essential as they incorporate both environmental and social costs. Thus, public values such as the land-related SDGs are part of the sustainable assessment and solution.

## 7. Conclusions

There is an increasing pressure on land, and due to improper use, land resources are quickly degrading, which will create even greater pressure on the remaining land. This calls for a new sustainable approach to land use and land management. There is a sense of urgency; the deadline for LDN (2030) is pressing, especially when it comes to environmental issues. Healthy soils and healthy land are essential to achieving many of the societal goals in the framework of the SDGs.

- To arrive at sustainable systems, broad and integrated approaches from an environmental, economic, and social point of view are needed, spanning the socio-ecological continuum of the systems that we wish to protect from degradation and manage sustainably.
- For the successful implementation and realization of the SDGs, a systems approach is necessary. The SGDs are not 17 separate goals that can be dealt with one by one. Instead, they should be seen as interlinked goals that can only be achieved through smart planning using the power of the natural and social system.
- The four approaches in this paper—systems thinking, connectivity, nature-based solutions, and a regenerative economy—are strongly interrelated. Systems thinking lies at the base of the three others, stressing not only feedback loops but also delayed responses. Their simultaneous use will result in more robust solutions, which are sustainable from an environmental, societal, and economic point of view. Short-term management needs to be embedded in long-term landscape vision and planning.
- Paradigm shifts are needed: to move from environmental protection to sustainable use and management and from a dominant economic and function-driven approach towards a natural system-based approach. To accomplish this, new business models are needed; an approach that integrates environmental, social, and economic interests. Only by making the transition

towards integrated solutions based on a socio-ecological systems analysis using concepts such as nature-based solutions do we stand a chance to achieve Land Degradation Neutrality by 2030.

**Author Contributions:** Conceptualization: all authors; Methodology, S.K. and G.M.; Investigation, S.K., G.M., M.d.C., C.M., S.V.; Writing-Original Draft Preparation: all authors; Writing-Review & Editing, all authors.

**Funding:** This research received no external funding.

**Conflicts of Interest:** The authors declare no conflicts of interest.

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
