# Peer review of "Soil-Related Sustainable Development Goals: Four Concepts to Make Land Degradation Neutrality and Restoration Work"

_land, doi:10.3390/land7040133_

Round 1
Reviewer 1 Report
This is an interesting manuscript exploring the tricky issue of how to implement sustainability. I personally find it opportune and on time, in view of the LDN initiative and the multiple searches around it. The language is very good, as well as the general presentation of the message, especially considering that this is more an essay than a method-based manuscript. I recommend it to be published short of some minor changes listed below:
The quality of the figures is bad, at least in the copy provided for review. It is difficult to read small coloured letters, and for that reason I could not assess completely the meaning in some cases.
Section 2.1: Because the presentation is focused on soils, some statements referring to land degradation (e.g. line 94) would improve by changing to soil degradation. In general, this section should be edited to refer to soil degradation.
Section 2.2, line105: The concept of eutrophication is only applied to water bodies. Please rephrase.
Example 1: Whilst it is essentially true that agricultural intensification and drip irrigation has encroached land degradation, this example is so generic that it should be better referred to Western Mediterranean. The crops associated with such orchards are not described. There is no such 'millennia old farming system' in Spain or any other Mediterranean country. On the contrary, the Mediterranean in general lacks of reference states because it has been in permanent change and has evolved with human populations. In the case of Spain, several Global Change events ('reconquista' in XV, 'desamortizaciones' from XVIII to XIX, mining and associated deforestation in XIX, rural peak population at the end of XIX, the entry in the EU in XX) have made frequent and dramatic changes in land management. The statement that no-tillage increases sediment yield (lines 441-443) should be further justified, not simply referenced. In general, the authors simplify the example to a point that is unconsistent with their claim to account for complexity. A constructive suggestion would be to address only the relatively recent trend towards intensification and short-term profit in Western European Mediterranean countries, as there is nothing specifically Spanish in this example.
Example 2: Similarly to the previous example, this text is richer in statements than in mechanisms. The case study orientation of this example would lend to describe precisely flows and feedbacks in the terms of ecological and socio-economic components dealing with sustainability. However, the reader is presented with simplified and idyllic visions of a restored landscape, of which not even the location is known. Further to that, the scale differences between this and the previous example (region or country vs. small case study) should be discussed, as well as their implications to apply the proposed paradigm.
Example 3: At least, this example starts by a concrete storyline and it is later discussed in the concrete terms of a sustainability component. This example is well developed and should be an example for the two previous ones, if you let me the game word.
Author Response
Reviewer 1:
This is an interesting manuscript exploring the tricky issue of how to implement sustainability. I personally find it opportune and on time, in view of the LDN initiative and the multiple searches around it. The language is very good, as well as the general presentation of the message, especially considering that this is more an essay than a method-based manuscript. I recommend it to be published short of some minor changes listed below:
Response: Many thanks for your kind words and compliments.
The quality of the figures is bad, at least in the copy provided for review. It is difficult to read small coloured letters, and for that reason I could not assess completely the meaning in some cases.
Response: I guess this was due to the way it was send to you, the original figures are of high quality, partially made by a professional graphical designer.
Section 2.1: Because the presentation is focused on soils, some statements referring to land degradation (e.g. line 94) would improve by changing to soil degradation. In general, this section should be edited to refer to soil degradation.
Response: many thanks for this feedback. We do see your point, however, we would like to place the soil processes in a larger context. We purposely send the paper to LAND for this also. Therefore, if the editor allows we would like to keep the text as it is.
Section 2.2, line105: The concept of eutrophication is only applied to water bodies. Please rephrase.
Response: In our sentence we do refer to soils and water. We do not understand how to rephrase to improve this sentence.
Example 1: Whilst it is essentially true that agricultural intensification and drip irrigation has encroached land degradation, this example is so generic that it should be better referred to Western Mediterranean. The crops associated with such orchards are not described. There is no such 'millennia old farming system' in Spain or any other Mediterranean country. On the contrary, the Mediterranean in general lacks of reference states because it has been in permanent change and has evolved with human populations. In the case of Spain, several Global Change events ('reconquista' in XV, 'desamortizaciones' from XVIII to XIX, mining and associated deforestation in XIX, rural peak population at the end of XIX, the entry in the EU in XX) have made frequent and dramatic changes in land management. The statement that no-tillage increases sediment yield (lines 441-443) should be further justified, not simply referenced. In general, the authors simplify the example to a point that is unconsistent with their claim to account for complexity. A constructive suggestion would be to address only the relatively recent trend towards intensification and short-term profit in Western European Mediterranean countries, as there is nothing specifically Spanish in this example.
Response: Thanks for the constructive feedback. We have changed on the one hand to make this example more generic to be exemplary for all Western Mediterranean countries. Although this is supposed to be an example, and not necessarily true everywhere. On the other hand, we tried to make it more specific to explain the crops involved, and explain our claims for no-tillage with herbicides as more unsustainable then tillage as was done for Millenia (without using chemicals). We do not claim that agriculture in Spain didn’t change before, but the fields were always ploughed, during all the periods that are identified by the reviewer. Only with the introduction of herbicides this practice has been abandoned on a large scale. Of course there are now also organic farmers that use cover crops, but they are still rare. And indeed, large scale, unsustainable intensive farming systems are more and more the standard in countries like Spain.
Example 2: Similarly to the previous example, this text is richer in statements than in mechanisms. The case study orientation of this example would lend to describe precisely flows and feedbacks in the terms of ecological and socio-economic components dealing with sustainability. However, the reader is presented with simplified and idyllic visions of a restored landscape, of which not even the location is known. Further to that, the scale differences between this and the previous example (region or country vs. small case study) should be discussed, as well as their implications to apply the proposed paradigm.
Response: We have added the location of the site where this project was done. The Project was at the Andalusian Altiplano Estepario, an area of 630,000 hectares, with 250 local stakeholders involved. We have presented this case study as a regional approach, unlike case 1 which is more a farm-scale approach, and in the discussion (section 6) we have elaborated further on the scale issue.
Example 3: At least, this example starts by a concrete storyline and it is later discussed in the concrete terms of a sustainability component. This example is well developed and should be an example for the two previous ones, if you let me the game word.
Response: thank you for your compliments, we tried to use the comprehensiveness of this example for the two others also.
Reviewer 2 Report
A very good article about contemporary topics. Combining the domains of formal regulations, economy, society and biosphere its a really key to sustainable land use. The considerations were supported by very well selected and elaborated examples.
Thank you especially for drawing attention to the imprecise basis for determining soil degradation. I'm still talking about it that the indicators, mainly based on chemical composition and water erosion threat, are not enough to proper determination the condition of the soil.
Detailed comments:
In the abstract, please add a brief note that the example considerations concern agricultural areas.
In my opinion, all texts in tables (except headers) should be aligned to the left - it looks better.
226- ; A factor that complicates the situation may also be the multi-functionality of the area (described in the article) - for example, in typical urban settlements, soil degradation is rarely adjudicated. This formal step would cause problems that will be difficult to overcome.
400; could have - yes, but not always have, mainly because of the developers' pressure
Please consider the addition of information contained in the elaboration "Urbanization : Challenge and Opportunity for Soil Functions and Ecosystem Services : Proceedings of the 9th SUITMA Congress" (Springer).
Author Response
Reviewer 2
A very good article about contemporary topics. Combining the domains of formal regulations, economy, society and biosphere its a really key to sustainable land use. The considerations were supported by very well selected and elaborated examples.
Response: Many thanks for your compliments.
Thank you especially for drawing attention to the imprecise basis for determining soil degradation. I'm still talking about it that the indicators, mainly based on chemical composition and water erosion threat, are not enough to proper determination the condition of the soil.
Response: Indeed, this is a problem that we as the soil science community need to look into more closely. In this paper we only briefly touch upon this, but in future papers we hope to elaborate more on this.
Detailed comments:
In the abstract, please add a brief note that the example considerations concern agricultural areas.
Response: thanks for your suggestion, we added this to the abstract.
In my opinion, all texts in tables (except headers) should be aligned to the left - it looks better.
Response: thanks for your suggestion, we have adapted the tables according to your suggestion.
226- ; A factor that complicates the situation may also be the multi-functionality of the area (described in the article) - for example, in typical urban settlements, soil degradation is rarely adjudicated. This formal step would cause problems that will be difficult to overcome.
Response: Indeed, this is a problem. Soils are not considered as something to take into account in many decision-making processes. With papers like this one, we hope to make soils more visible and make decision makers and the general public as a whole more aware of the importance of soils for any area. Including urban areas.
400; could have - yes, but not always have, mainly because of the developers' pressure
Response: Indeed, we agree with the reviewers remark, but hope the work of Mazzucato and our reference to it, will raise the awareness of government officials of responsibilities and possibilities.
Please consider the addition of information contained in the elaboration "Urbanization : Challenge and Opportunity for Soil Functions and Ecosystem Services : Proceedings of the 9th SUITMA Congress" (Springer).
Response: Many thanks for this suggestion. We have carefully considered this suggestion, however, as the scope of this paper lies with sustainable land use in non-urban areas, we would prefer not to take this reference up in our paper.